# CRNNTL: Convolutional Recurrent Neural Network and Transfer Learning for QSAR Modeling in Organic Drug and Material Discovery

**DOI:** 10.3390/molecules26237257

**Published:** 2021-11-30

**Authors:** Yaqin Li, Yongjin Xu, Yi Yu

**Affiliations:** 1West China Tianfu Hospital, Sichuan University, Chengdu 610041, China; 2Department of Chemistry and Molecular Biology, University of Gothenburg, Kemivägen 10, 41296 Gothenburg, Sweden; yongjin.xu@chem.gu.se

**Keywords:** DEEP learning, molecular autoencoders, QSAR, RNN, CNN, transfer learning

## Abstract

Molecular latent representations, derived from autoencoders (AEs), have been widely used for drug or material discovery over the past couple of years. In particular, a variety of machine learning methods based on latent representations have shown excellent performance on quantitative structure–activity relationship (QSAR) modeling. However, the sequence feature of them has not been considered in most cases. In addition, data scarcity is still the main obstacle for deep learning strategies, especially for bioactivity datasets. In this study, we propose the convolutional recurrent neural network and transfer learning (CRNNTL) method inspired by the applications of polyphonic sound detection and electrocardiogram classification. Our model takes advantage of both convolutional and recurrent neural networks for feature extraction, as well as the data augmentation method. According to QSAR modeling on 27 datasets, CRNNTL can outperform or compete with state-of-art methods in both drug and material properties. In addition, the performances on one isomers-based dataset indicate that its excellent performance results from the improved ability in global feature extraction when the ability of the local one is maintained. Then, the transfer learning results show that CRNNTL can overcome data scarcity when choosing relative source datasets. Finally, the high versatility of our model is shown by using different latent representations as inputs from other types of AEs.

## 1. Introduction

For the excavation of crucial molecular factors on properties and activities, quantitative structure-activity relationship (QSAR) has been an active research area in the past 50+ years. In QSAR, the molecular representations (or descriptors), as the input features of the modeling, represent chemical information of actual entities in computer-understandable numbers [1,2,3]. Historically, molecular fingerprints, such as extended-connectivity fingerprints (ECFPs), have been widely used as representations for the modeling in drug and material discovery [4]. Recent development in deep neural network facilitates the utilization of different molecular representations such as latent representations [5,6]. Latent representations are fixed-length continuous vectors that are derived from autoencoders (AEs) with encoder−decoder architecture [7]. Although AEs are shown as a generative algorithm for de novo design studies in the beginning [8,9], latent representations from encoders of AEs have been extracted for QSAR modeling [6]. Without sophisticated human-engineered feature selection, these kinds of representations show competitive performance compared with traditional ones [5]. Furthermore, the representations and QSAR models can be used for multi-objective molecular optimization to tackle inverse QSAR problem [10].

To obtain latent representations, sequenced-based strings, such as SMILES (Simplified Molecular Input Line Entry Specification), are employed as inputs into AEs. The method in detail is described in the next section. In recent years, great interest has been aroused by developing QSAR models based on latent representations. One of the earliest achievements in this field was the chemical variational autoencoders (VAE) by Aspuru-Guzik et al. [7]. After generating latent representations for the encoder, two fully connected artificial neural networks (ANNs) were used for the prediction of water−octanol partition coefficient (logP), the synthetic accessibility score, and drug-likeness. More recently, Winter et al. have shown that latent representation applied on support vector machine (SVM) outperforms ECFPs and graph-convolution method by the translation AE model (named CDDD) [6]. Subsequently, a convolutional neural network (CNN) was trained for the QSAR prediction after generating latent representations [11,12]. Owing to the architectural characteristic of local connectivity, shared weight, and pooling [13], some studies in the literature indicated that CNNs perform better in QSAR modeling compared with ANN and other traditional models [14].

On the one hand, latent representations should have sequence features if they are derived from sequence-based strings [15]. As far as we know, recurrent neural network (RNN) outperforms other models including CNN when dealing with sequence data in some areas, such as natural language processing (NLP) [16] and electrocardiogram classification (ECG) [17]. Even though CNN performs well due to its ability of local feature selection, RNN has its advantage in global feature discovery [18]. With regard to molecules, it has similarity with the NLP and ECG. We assume that the local feature represents the type of atoms and functional group, and global feature is the atomic arrangement. Molecular properties and activities depend on not only atom types and functional groups (local features) but also the arrangement (global features). In other words, the sequence of the atoms or functional groups plays an important role in molecular properties. However, very few works have been done to study the modeling performance with the difference of the molecular sequence (global features).

On the other hand, the scarce availability of labeled data is the major obstacle for QSAR modeling [2]. According to the probably approximately correct theory, the size of training data plays a key role in the accuracy of machine learning methods [19]. Nonetheless, the available datasets are small at most stages of the QSAR pipeline, especially bioactivity modeling. One strategy to solve this problem is transfer learning algorithms [20]. Transfer learning is one kind of machine learning method that takes advantage of existing, generalizable knowledge from other sources [21]. For example, Li et al. reported an RNN model pretrained on one million unlabeled molecules from ChEMBL and fine-tuned with Lipophilicity (LogP), HIV, and FreeSolv data [22]. It indicated that transfer learning improved the performance strongly compared with learning from scratch. Additionally, Iovanac et al. improve the QSAR prediction ability by the integration of experimentally available pKa data and DFT-based characterizations of the (de)protonation free energy [23]. However, few transfer learning strategies for multiple bioactivity and material datasets have been presented. In comparison with physicochemical and physiological properties, the measurements of molecular bioactivities are more time and resource consuming. Additionally, the scales of the datasets about experimental material properties are usually less than 1000 [24]. Therefore, it is advisable to train more transfer learning models among this type of dataset. In this way, the information from other datasets would be transferred into small ones to facilitate molecular design and drug discovery.

Herein, we describe the convolutional recurrent neural network and transfer learning (CRNNTL) method to tackle the problems with molecular sequence and data scarcity. The convolutional recurrent neural network (CRNN) is loosely inspired by the architectures proposed in the applications of polyphonic sound detection and electrocardiogram classification [25,26,27], which integrates the advantages of CNN and RNN (gated recurrent units used here, named GRU) at the same time. The concept is illustrated in Figure 1 in which the model is constituted by the CNN, GRU, and dense layer parts. Firstly, the CRNN model is tested using diverse benchmark datasets including drug and material properties and compared with CNN and classical methods, such as random forest (RF) and SVM. Then, an isomers-based dataset is trained by CRNN and CNN to elucidate the improved ability of CRNN in the global feature learning. Next, we demonstrated that the transfer learning part of CRNNTL could be used to improve the performance for scarce bioactivity and material datasets. Finally, the high versatility of CRNNTL was shown by QSAR modeling based on two other latent representations derived from different types of AEs. 

## 2. Results and Discussion

### 2.1. Model Optimization

The performance of the deep learning method depends on the hyperparameters and network architecture. Firstly, hyperparameter optimization was performed by grid search. A series of models were built using different combinations of batch size for the whole neural network, as well as optimizer learning rate and activation functions for the CNN or GRU parts, respectively. The hyperparameter settings are summarized in Table 1. The model provided the best performance with 128 batch size and ReLU activation function for the two parts, while the best learning rates were 0.0001 and 0.0005 for the CNN and GRU, respectively.

Then, architecture optimization was performed on both CNN and GRU parts. As for the CNN part, the model performed well with three convolutional layers. As the number of layers increases, little improvement or a slight decrease (less than 2%) occurred, as shown in Appendix A. This implied that deeper CNN network architecture involving more parameters would affect the final performance of the model, especially for the relatively small dataset in our cases [11]. Considering the increase in training time and computing cost, three convolutional layers was most suitable for the modeling. Regarding the GRU part, one bidirectional layer for the GRU was enough for model training. One of the reasons is similar with the one in the CNN case. Additionally, since the local feature has been extracted well by CNN, denser GRU layers are not needed any more.

### 2.2. CRNN for Drug Properties

After model optimization, the QSAR results by five models were compared with each other, including CNN, CRNN, and CRNN with the augmentation (AugCRNN), SVM, and RF. The inputs of the first four models are latent representation, while RF is based on ECFPs. The 20 datasets contain physicochemical properties, physiological properties, and bioactivities for drug-like molecules (detailed information in Materials and Methods). The random five-fold cross-validation was performed to compare the models with each other. Table 2 shows the results for regression datasets, while Table 3 demonstrates the results for classification tasks. Except for a few tasks, such as Lipo, the AugCRNN method yielded better results than any of the other ones applied on latent representation, as well as the classical machine learning method on ECFPs. In comparison to the CRNN method, the augmentation can overcome the limitation of small data size, which was found in the artificial intelligence area for not only drug discovery [28] but also other applications such as computer vision and natural language procession [29].

In addition, it should be noted that the CRNN provided higher area under the receiver characteristic curve (ROC-AUC) values or coefficients of determination (r^2^) in almost 20 tasks compared with the CNN method (except for MP and BACE), without data augmentation at the same time. As we mentioned above, the CRNN and CNN models have the same convolutional part and fully connected structures. Therefore, it suggested that the GRU in CRNN results in better performance in QSAR. In the research area of electrocardiogram (ECG) analysis, the local feature represents different kinds of amplitudes and intervals in a short time, while the global one represents the permutation and combination of them [27]. It is reported that the CRNN model can learn not only amplitudes and intervals but also their permutation and combination [26]. Given the good QSAR performance of our model, we hypothesize that CRNN can effectively extract not only the molecular local feature but also the global one in which the molecular local feature represents the type of atoms and functional group, and the global feature is the atomic arrangement.

Although three different learning algorithms were performed on each task, the modeling performance based on the ECFPs could be further improved by choosing best flavor of fingerprint representation. However, it would take considerable training time due to 20 tasks. Another thing we should pay attention to is that the performances of property prediction are always low for the AUR3, PI3, and MP datasets, whichever model was used. It might arise from the data scarcity of the three datasets. The prediction improvement would be realized by the transfer learning method and will be shown and discussed in a later section. To sum up, despite our harsh evaluation scheme, it still indicated that CRNN can compete or outperform other baseline methods for molecular QSAR modeling in various drug properties due to the abilities of both local and global feature extraction abilities. In the next section, the QSAR performance for material properties will be demonstrated and discussed. 

### 2.3. CRNN for Material Properties

We turned our attention to the ability of material property prediction by our method. The same five QSAR models were studied based on seven datasets. The first five of the datasets belong to the traditional properties of luminescent materials (including absorption peak position, emission peak position, extinction coefficient in logarithm, bandwidth in full width at half maximum, and the lifetime). The sixth one is the triplet state energy (E_T1_) for the thermally activated delayed fluorescent (TADF) molecules, and the last one is the power conversion efficiency for organic solar cells from HOPV15 [30]. The detailed information for the datasets and evaluation method are shown in the Method section. 

Table 4 shows the result for seven datasets. As for the luminescent properties, AugCRNN outperformed other models in most cases, and the performance of CRNN is always better than that of CNN, which is consistent with the result in drug property prediction. It should be noted that all the performances of AugCRNN are lower than in the previous work using the Deep4Chem interface. One reason is that a random five-fold cross-validation method was used here, while the previous work only split the data into train, validation, and test sets. The other one is that a relatively smaller dataset is available online for training. In short, CRNN is still an efficient QSAR method for the property prediction of luminescent materials.

E_T1_ is the one of the most important values for TADF molecules in the application of organic light-emitting diodes, since the reverse intersystem crossing occurs from the triplet state to the singlet state to facilitate the energy transfer process [31]. In addition, PCE is a vital value as well to show the energy transfer performance of the solar cell [24]. As far as we know, it is the first time that QSAR modeling for the E_T1_ was studied based on TADF molecules. As for the PCE modeling, previous work shows excellent performance when using microscopic properties as the representation. However, this kind of representation is expensive to compute or experimentally determine. Except for the data scarcity, the low performance for the PCE might result from the unsuitability of our AE. Nowadays, most AEs are designed and trained for drug-like molecules [5,6,27]. Molecules for photovoltaic materials have much larger molecular weight, LogP, and more conjugated structures. It is anticipated that more proper AE designed and trained for photovoltaic molecules would result in better QSAR performance in PCE in the future. Given the above, our method showed the competitive modeling performance of various material properties without sophisticated molecular representation computation. We will next study why CRNN can perform better than CNN in most datasets.

### 2.4. CRNN for the Isomers-Based Dataset

So far, the difference between the local and global feature of molecules has been demonstrated. It is well-acknowledged that CNN could extract the local feature effectively in molecular modeling. However, few works have been done for the molecular global feature extraction. One question in mind is which model will perform best on a dataset with only global feature variation. In chemistry, the isomers have the same atoms or functional groups (local feature) but different atomic arrangement (global feature). It could be the most coincident case for the study of the ability of global feature extraction. We will now explore the melting point QSAR modeling based on isomers by CNN, CRNN, and SVM applied on latent representation. The isomer information is extracted from PubChem and ChemSpider, which includes three types of isomers alcohols and ethers, amino acids and amides, as well as carboxylic acids and esters. Some cases for three types of isomers are shown in Table 5.

As shown in Table 5, the melting points between isomers are different from each other, while the SMILES are made up of the same characters with different arrangements. For instance, the melting point of amino acids is much higher compared with the amides, since there is an internal transfer of a hydrogen ion from the -COOH group to the -NH_2_ group to leave an ion with both a negative charge and a positive charge. These ionic attractions take more energy to break, so the amino acids have high melting points, while this effect will not occur in amides. The two other types of isomers have different melting points due to a similar reason.

The r^2^ values of the isomers-based datasets are 0.81, 0.88, and 0.86 using CNN, CRNN, and SVM, respectively. In other words, the CRNN outperforms CNN and SVM on the QSAR modeling applied on the dataset with only difference in the global feature. The relatively huge different performance (10%) between CRNN and CNN strengthened our hypothesis that CRNN has better performance in global feature extraction when the ability of local feature learning is maintained. Figure 1 demonstrates the architecture of CRNN and its abilities in two types of feature learning. At first, the convolutional part extracted the information of the types of atoms and functional groups. Then, the knowledge about the atomic arrangement was learned by the GRU part. Finally, all the features come into dense layers for classification or regression. It should be noted that in regular melting point modeling among the 20 datasets above, the r^2^ by the CRNN model was lower than the one by CNN. To elucidate the contradiction, the molecule structures in the dataset above were checked. It turned out that no isomer in the dataset was found. Therefore, the global feature extraction might not be the key ability for this QSAR modeling. Moreover, since CRNN has many more parameters for fitting in the training process, a small dataset (283 molecules) might contribute to under-fitting for it. Therefore, these seemingly contradictory results between two MP datasets have been explained. We will next show the transfer learning performance by CRNNTL.

### 2.5. Transfer Learning for Small Datasets of Drug Properties

Some of the 20 datasets above were used to investigate the knowledge transfer ability of our model. In detail, MP, SIDER, AUR3c, and PI3 were considered as target datasets, while BP, TOX21, FGFR1, MTOR, and EGFR acted as source datasets. Regarding the transfer learning for physiology and physicochemical datasets, the ROC-AUC and r^2^ results are reported in Appendix A in detail. With the big datasets acting as a source target, the transfer learning performance achieved about a 5% improvement in scores (for both MP and SIDER) compared to learning from scratch, which is consistent with the previous work due to the knowledge transferred from the big dataset into the small one [32].

Having settled the transfer learning performance of CRNNTL in the physiological and physicochemical datasets, we turned our attention to the knowledge transfer ability to bioactivities. As shown in the first part, the regression results of AUR3 and PI3 were relatively worse than the other ones because of the small data size (shown in Table 2). Hence, larger datasets, including FGFR1, MTOR, and EGFR, were used to transfer the knowledge into AUR3 and PI3. Figure 2 showed the results of the transfer learning based on the CRNN method. On the one hand, when FGFR1 serves as the source dataset, the r^2^ of both PI3 and AUR3 increased by about 10% compared to learning from scratch. On the other hand, regarding the MTOR, QSAR for PI3 achieved almost a 30% improvement in r^2^. However, no improvement was realized in case of AUR3. Despite the adequate QSAR performance and size of the EGFR dataset, there was no statistical improvement when EGFR acted as a source for both the PI3 and AUR3 targets. Therefore, it demonstrated that the transfer learning ability did not only depend on the performance and size of the source dataset. 

According to the lock–key model in pharmaceutical science, the protein structures play a significant role in the ligand–target interaction as well as the molecular ones, which is different from the cases in physicochemical and physiological properties. Within the target structures, local binding site similarities could be more important than global similarities [33,34]. Hence, the local binding site similarities were compared using SMAP *p*-values among the targets mentioned above [35]. SMAP *p*-values represent the binding site similarity between two targets, which is based on a sensitive and robust ligand binding site comparison algorithm [36,37,38]. The lower the SMAP *p*-value, the more similarity between binding sites.

As shown in Table 6, in comparison with MTOR, the SMAP *p*-values of PI3 is 7.8 × 10^−6^, which is the lowest among others, while the *p*-value between MTOR and AUR3 implied that there is insignificant similarity between them. These results indicated that a high binding site similarity resulted in the efficient knowledge transfer from the source dataset into the target dataset. When FGFR1 acted as a source dataset, the results were consistent with the ones above: due to moderate binding site similarities with the source, moderate improvements were achieved for PI3 and AUR3 during the knowledge transfer process. As for EGFR, no statistical improvement in two cases arose from little binding site similarities with both two targets. In brief, a strong positive correlation between transfer learning ability and binding site similarity was found. 

The transfer learning performance by CNN was also studied and shown in Appendix A. Given the binding site similarities, high improvement was achieved as well. However, the result of CRNN for transfer learning was still better than that of the CNN method. To elucidate the difference, we tried to freeze the weights either in the local (convolutional) or global (GRU) feature learning part when fine-tuning the models. As shown in Appendix A, similar improvement was shown when freezing the weights in the local feature learning part, while there was little improvement as weights in the global part frozen. It indicated that fine tuning in the GRU part played a key role in the transfer learning process. That is to say, local features can be shared between different targets with binding site similarity in CRNNTL. Then, valuable local features can be transferred from the source dataset. Finally, modeling performance was improved efficiently by fine tuning the global feature learning (GRU) part.

### 2.6. Transfer Learning for Small Datasets of Material Properties

Compared with datasets of luminescent properties, the sizes of datasets of E_T_ and PCE are small. Inspired by the knowledge transfer between drug properties, five luminescent datasets acted as sources, while E_T_ and PCE were considered as target datasets. Regarding E_T_, a 12% improvement was achieved when Em_max_ was used as a source dataset. Additionally, the r^2^ increased by 9% when Abs_max_ was used as the source. In contrast, no matter which source was used, no improvement was shown compared with the learning from scratch method. 

The result above might be explained by analyzing the property relationships between them. Em_max_ and Abs_max_ represent the information of the energy level of a molecular excited singlet state (E_T1_). According to a simple two-electron two-state model, E_S1_ and E_T1_ can be written as [39]
E_S__1_ = h_H_ + h_L_ + J_HL_ + K_HL_,(1)
E_T__1_ = h_H_ + h_L_ + J_HL_ − K_HL_.(2)

Here, h_H_ and h_L_ represent the one-electron energy of the HOMO and LUMO orbital, respectively, while J_HL_ is the Coulomb repulsion energy between electron 1 on the HOMO and electron 2 on the LUMO, and K_HL_ denotes the corresponding electron exchange energy. Hence, the difference between the E_S1_ and E_T1_ is small. The knowledge from the Em_max_ and Abs_max_ datasets can be effectively transferred for the QSAR modeling of E_T1_. As for PCE, the value can be expressed as [24]
(3)$$\mathrm{PCE}=100\times \frac{V_{oc}\times \mathrm{FF}\times J_{SC}}{P_{in}},$$
where V_OC_, FF, J_SC_, and P_in_ denote the open-circuit voltage, fill factor, short-circuit current, and input power, respectively. Even though the five luminescent properties are relevant to the values mentioned above, they do not play key roles for the PCE. Therefore, transfer learning is inefficient between the source and target. In a word, efficient transfer learning could be realized by CRNNTL when a relevant source and target dataset are selected.

### 2.7. Versatility

To demonstrate the versatility of our strategy, two latent representations were generated from one AAE [40] and another VAE [9] (named DDC). Then, their QSAR performance were studied based on CRNN and other baseline methods as well as transfer learning for the modeling improvement on a small dataset. As shown in the Appendix A, the results were accordant with the aforementioned ones derived from CDDD VAE: firstly, AugCRNN outperformed other models among most datasets; secondly, the results using CRNN were better than the one by CNN in most cases due to the abilities of both global and local feature extractions; thirdly, up to 30% improvements achieved by the transfer learning module when taking binding site similarities into con-sideration. Meanwhile, the QSAR modeling performance based on the AAE was not as good as the results of two other VAEs. Compared with CDDD VAE, fewer molecules were trained in the sequence-to-sequence model. Additionally, despite a similar amount of training data, SMILES enumeration was used to improve the performance in another DDC encoder. The results of AAE strengthen our hypothesis in material QSAR that the unsuitability of the AE would affect the QSAR performance. Given the above, our approach has high versatility to be used as QSAR modeling and transfer learning for other latent representations from different AEs.

## 3. Materials and Methods

### 3.1. Input

While latent representation could be generated from various initial molecular representation by encoders of AEs, in this work, we concentrated on the sequence-based SMILES as the input representations. The SMILES represent molecular structures where atoms are labeled as nodes and bonds between atoms are encoded as edges [41]. In the beginning, we used canonical SMILES to generate latent representation. Then, data augmentation for the training set was performed to optimize our modeling in which 9 other SMILES were randomly generated from the canonical SMILES for model training. Meanwhile, the testing set remained the same with the QSAR method without data augmentation.

### 3.2. AEs

Subsequently, the SMILES strings were encoded by AEs to generate fixed-length vectors as the latent representations. In recent years, developing AEs for molecular de novo design and QSAR modeling has become a hot topic for drug discovery [7,8]. In an AE, the latent representation is derived from the encoder. Then, the decoder part of an AE could be used to reconstruct the molecular structure. For instance, the VAE is one kind of developed AEs in which new samples could be generated by the decoders. Additionally, adversarial autoencoders (AAEs) are a modification of VAEs where an AE is combined with a generative adversarial network (GAN). Due to a prior distribution of the training, AAEs facilitate the generation of novel structures. In this study, two VAEs [6,9] and one AAE [40] were used for latent representation generation. Their description is provided in Appendix A. At first, based on the latent representation from CDDD [6] (one of the VAEs algorithm), we compare the CRNNTL performance on the QSAR modeling and transfer learning with state-of-the-art methods. Then, the versatility was studied based on the latent representation generated from the other VAE and AAE methods.

### 3.3. Datasets and Preprocessing

As for the QSAR in drug properties, the datasets include various physicochemical or physiological properties and bioactivities in which 10 for classification and 10 for regression were selected from different sources. Table 7 summarized the information about 20 datasets. They were obtained from DeepChem or other sources. The isomers-based dataset represents different melting point for 70 molecules. In the dataset, for some amino acids, it is difficult to get the exact melting point because they tend to decompose before melting. In such a case, the decomposition temperatures are used as labeling data instead. Each isomer couples were in the same group for cross-validation.

With regard to the dataset of material properties, the five datasets featuring luminescent properties were obtained from the previous work [52]. Meanwhile, the dataset of TADF molecules was collected from two reviews [31,53]. In addition, the dataset of molecules for solar cells is from HOPV15 [30]. Table 8 summarizes the information about 7 datasets.

### 3.4. The Architecture of QSAR Models

The architecture of CRNN has CNN, GRU, and dense layer parts, as shown in Figure 1. This new method was benchmarked against state-of-the-art ones, including CNN and SVM applied on latent representation as well as classical machine learning on ECFPs.

The hyperparameter settings were fixed based on the HIV and EGFR datasets, while other datasets were solely used for evaluating the final model. As for the CNN architecture, it contains convolutional and classification (or regression) parts. The local feature learning part has 3 convolutional layers. According to the literature [11,12,25,26], the kernel sizes of the convolution layers were 5, 2, and 5. Given the hyperparameter search, the numbers of the produced filters were 15, 30, and 60, respectively. The kernel sizes of the convolution were 5, 2, and 5, and the numbers of the produced filters were 15, 30, and 60, respectively. Batch normalization was used after each convolutional layer. After the pooling operation, the data went through two fully connected layers as the classification (or regression) part for which the output layer consisted of two neurons for classification tasks (or one neuron for regression). The hyperparameter and architecture optimization are shown in the next section. Early stopping was performed to avoid overfitting. More specifically, the r^2^ of the test dataset was monitored. When the r^2^ decreases more than 2 times, the stopping is triggered (the threshold is 0.005 or 0.01). The SVM modeling applied on latent representation was analyzed according to the previous work. Meanwhile, Random Forest (RF) was implemented in scikit-learn for the modeling based on ECFP.

### 3.5. Training and Evaluation

The end-to-end from scratch method was used for CNN training. As for the CRNN, the training for some datasets encountered no convergence. Referring to the literature about CRNN modeling for electrocardiogram classification [26], 3-phase protocol was used. In phase 1, the GRU part was blocked and the weight in the CNN and dense layers was updated. Then, the GRU part was unfrozen, and other parts were fixed. Finally, 3 parts were trained jointly. Meanwhile, if the size of the dataset is small, the batch size can be less than 128. Meanwhile, if the size of the dataset is equal to or larger than 300, the batch size is 128, which shows the best performance in grid search. When the batch size is smaller than 300 and larger than 100, the batch size is 64. When the batch size is equal to or smaller than 100, the batch size is 24.

The area under the receiver characteristic curve (ROC-AUC) values and coefficients of determination (r^2^) were used for the classification and regression tasks, respectively. Except for the ET and PCE datasets, others were randomly split into a training and test set using the sklearn package. Then, the 5-fold cross-validation was performed to compare with the performance by the aforementioned methods in which the testing dataset was kept apart from the hyperparameter fine tuning to avoid potential overfitting. Due to the small size of the datasets, ET and PCE modeling showed large standard mean errors compared with the r^2^. In order to ensure the test set lays within the domain of applicability of the model, a k-mean clustering algorithm was used to divide the datasets into the training (80%) and test (20%) sets [54].

### 3.6. Transfer Learning

The source dataset was first trained using the CRNN model by 5-fold cross-validation. Then, the best performance model was selected and saved for the transfer learning use. Three transfer learning methods were used here. The first one is based on the conventional method. The pretrained model was loaded. After loading the target data, the convolutional part was frozen, and other parts were trained. Finally, the convolutional part was unfrozen, and the whole network was fine-tuned. The second way follows the same process in the beginning. However, the difference was that there is no unfreezing for the convolutional part. In other words, the local feature learning part was not tuned in the knowledge transfer process. As for the third method, the GRU part was frozen in the whole training process.

## 4. Conclusions

We demonstrate a convolutional and recurrent neural network and transfer learning model (CRNNTL) for both QSAR modeling and knowledge transfer study. As for the CRNN part, it integrates the advantages of both convolutional and recurrent neural networks as well as the data augmentation method. Our method outperforms or competes with the state-of-art ones in 27 datasets of drug or material properties. We hypothesize that the excellent performance results from improvement of the ability of global feature (atomic arrangement) extraction by the GRU part. Meanwhile, the ability of local feature (types of atoms and functional groups) extraction is maintained by the CNN part. This assumption was strengthened by training and testing isomer-based datasets. Even though more parameters need to be trained when the data size is not large, the performance of CRNN is almost 10% higher than that of traditional CNN because of the outstanding ability of global feature learning. In drug and material discovery, high-proportioned isomers and derivatives need to be tested for their properties. Therefore, our model provides a new strategy for molecular QSAR modeling in various properties.

With regard to the transfer learning part, effective knowledge transfer from a larger dataset into a small one can occur for QSAR modeling in both drug and material properties. When considering binding site similarities, up to 30% improvement was achieved by CRNNTL between different bioactivity datasets. In addition, a considerable increase was shown for the transfer learning between the larger dataset for luminescent molecules and the smaller one for OLED materials. Accordingly, transfer learning could facilitate the QSAR performance when considering the correlation between datasets. Hence, CRNNTL could be a potential method to overcome the data scarcity for QSAR modeling.

At last, CRNNTL showed high versatility by testing the model on different latent representations from other types of autoencoders. We anticipate that the performance of QSAR modeling by our method could be further improved by the latent representation generated from the AEs, which is more suitable for molecules in material science. Accordingly, we expect that CRNNTL could pave the pathway for drug and material discovery in the big data era.

## Figures and Tables

**Figure 1 molecules-26-07257-f001:**
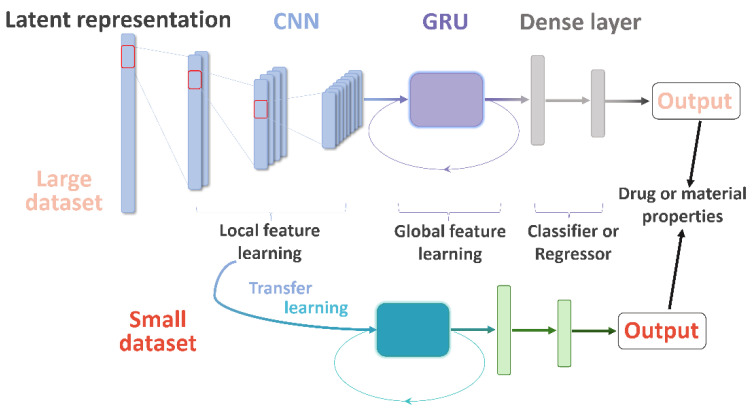
The architecture of CRNN and transfer learning method between large and small datasets.

**Figure 2 molecules-26-07257-f002:**
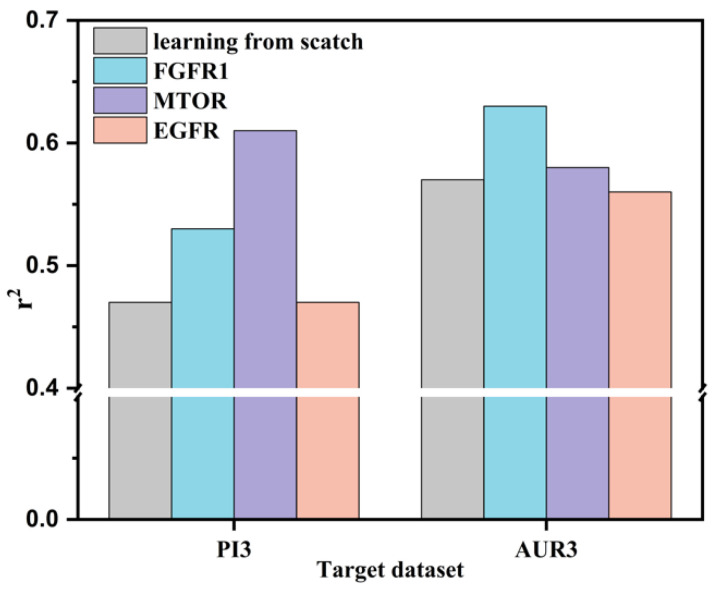
Transfer learning results for PI3 and AUR3 as target datasets and FGFR1, MOTR, and EGFR as source datasets, learning from scratch as the baseline to evaluate the improvement.

**Table 1 molecules-26-07257-t001:** Overview of the optimization settings.

Settings	CNN	GRU
activation function	(anh, ReLU)	(Sigmoid, ReLU)
learning rate	(0.001, 0.0005, 0.0001)	(0.001, 0.0005, 0.0001)
number of layers	(3–5)	(1,2)

**Table 2 molecules-26-07257-t002:** Coefficient of determination (r^2^) for regression datasets of drug properties.

Dataset ^a^	CNN	CRNN	AugCRNN	SVM	RF ^b^
EGFR	0.67	0.70	**0.71**	0.70	0.69
EAR3	0.64	0.68	**0.70**	0.65	0.53
AUR3	0.55	0.57	**0.61**	0.60	0.54
FGFR1	0.63	0.68	**0.72**	0.71	0.68
MTOR	0.64	0.68	**0.70**	**0.70**	0.66
PI3	0.43	0.47	0.50	**0.52**	0.45
LogS	0.91	0.92	**0.93**	0.92	0.90
Lipo	0.63	0.67	0.70	**0.73**	0.66
BP	0.95	0.96	**0.97**	0.96	0.93
MP	0.47	0.46	**0.52**	0.46	0.45

The standard mean errors are shown in Appendix A. Bold texts represent the best performance. ^a^ The information in detail for each dataset is summarized in Materials and Methods. ^b^ Calculated with ECFP representation.

**Table 3 molecules-26-07257-t003:** The area under the receiver characteristic curve (ROC-AUC) for classification datasets of drug properties.

Dataset ^a^	CNN	CRNN	AugCRNN	SVM	RF ^b^
HIV	0.80	0.82	**0.83**	0.76	0.78
AMES	0.86	0.87	0.88	**0.89**	**0.89**
BACE	0.88	0.89	0.90	0.90	**0.91**
HERG	0.83	0.84	**0.86**	**0.86**	0.85
BBBP	0.88	0.89	0.91	**0.93**	0.89
BEETOX	0.89	0.91	**0.92**	**0.92**	0.90
JAK3	0.72	0.74	**0.77**	0.76	0.76
BioDeg	0.75	0.77	**0.78**	0.74	0.73
TOX21	0.75	0.77	**0.78**	0.74	0.73
SIDER	0.68	0.70	**0.72**	0.70	0.68

The standard mean errors are shown in Appendix A. Bold texts represent the best performance. ^a^ The information in detail for each dataset is summarized in Materials and Methods. ^b^ Calculated with ECFP representation.

**Table 4 molecules-26-07257-t004:** Coefficient of determination (r^2^) for regression datasets of material properties.

Dataset ^a^	CNN	CRNN	AugCRNN	SVM	RF ^b^
Abs_max_	0.75	0.87	**0.90**	0.89	0.88
Em_max_	0.72	0.82	**0.85**	**0.85**	0.81
Logε	0.52	0.73	**0.75**	0.73	0.73
σ_abs_	0.44	0.56	**0.59**	0.52	0.55
lifetime	0.45	0.59	**0.63**	0.59	0.58
E_T1_	0.49	0.48	**0.52**	0.48	**0.52**
PCE	0.42	0.43	**0.47**	0.42	0.43

The standard mean errors are shown in Appendix A. Bold texts represent the best performance. ^a^ The first five datasets represent the absorption peak position (Abs_max_), emission peak position (Em_max_), extinction coefficient in logarithm (logε), bandwidth in full width at half maximum (σ_abs_), and the molecular lifetime, respectively; the sixth one is the triplet state energy (E_T1_) for the TADF molecules; the last one is the power conversion efficiency (PCE) from HOPV15 ^b^ Calculated with ECFP representation.

**Table 5 molecules-26-07257-t005:** Names, SMILES, molecular structures, and melting points of isomers.

Name	SMILES	Molecular Structure	Melting Point (°C)
2-Hydroxypropanamide	CC(O)C(N)=O	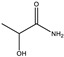	78
Alanine	CC(N)C(=O)O	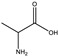	292 ^a^
1,3-Dimethoxypropane	COCCCOC	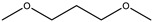	−82
1,5-Pentanediol	OCCCCCO	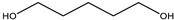	−16
Methyl benzoate	COC(=O)c1ccccc1	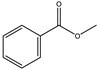	−12
Phenylacetic acid	O=C(O)Cc1ccccc1	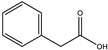	77

^a^ Alanine decomposes before melting, the value here is the temperature at which it decomposes.

**Table 6 molecules-26-07257-t006:** Binding site similarities between different targets. SMAP *p*-values represent the similarities and the lower the SMAP *p*-value, the more similarity between different targets.

Targets	PI3	AUR3
FGFR1	1.3 × 10^−4^	2.1 × 10^−5^
MTOR	7.8 × 10^−6^	9.4 × 10^−3^
EGFR	5.2 × 10^−3^	8.6 × 10^−3^

**Table 7 molecules-26-07257-t007:** Overview of the datasets of drug properties (left for regression and right for classification).

Acronym	Description	Size	Acronym	Description	Size
EGFR	Epidermal growth factor inhibition [42]	4113	HIV	Inhibition of HIV replication [43]	41101
EAR3	Ephrin type-A receptor 3 [42]	587	AMES	Mutagenicity [6]	6130
AUR3	Aurora kinase C [42]	1001	BACE	Human β-secretase 1 inhibitors [43]	1483
FGFR1	Fibroblast growth factor receptor [44]	4177	HERG	HERG inhibition [40]	3440
MTOR	Rapamycin target protein [45]	6995	BBBP	Blood–brain barrier penetration [43]	1879
PI3	PI3-kinase p110-gamma [46]	2995	BEETOX	Toxicity in honeybees [47]	188
LogS	Aqueous solubility [43]	1144	JAK3	Janus kinase 3 inhibitor [48]	868
Lipo	Lipophilicity [43]	3817	BioDeg	Biodegradability [49]	1698
BP	Boiling point [50]	12451	TOX21	In-vitro toxicity [43]	7785
MP	Melting point [51]	283	SIDER	Side Effect Resource [43]	1412

**Table 8 molecules-26-07257-t008:** Overview of the datasets of material properties.

Acronym	Description	Size
Abs_max_	absorption peak position [52]	6433
Em_max_	emission peak position [52]	6412
Logε	extinction coefficient in logarithm [52]	3848
σ_abs_	bandwidth in full width at half maximum [52]	1606
lifetime	molecular lifetime [52]	2755
E_T1_	triplet state energy [31,53]	60
PCE	power conversion efficiency [30]	249

## Data Availability

The code and data are available on https://github.com/YiYuDL/CRNNTL; accessed on 5 November 2021.

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
