# Peer review of "CRNNTL: Convolutional Recurrent Neural Network and Transfer Learning for QSAR Modeling in Organic Drug and Material Discovery"

_molecules, 2021, doi:10.3390/molecules26237257_

Round 1

Reviewer 1 Report

The paper demonstrates that neural networks with convolutional and recurrent layers, trained on autoencoder's latent representations, can compete with, and even outperform baseline models, such as regular CNN, SVM or RF, on QSAR modeling of drug and material properties. Furthermore, models enhanced through transfer learning successfully mitigate the problem of small datasets.

Overall, the opinion of this reviewer is that the paper provides adequate evidence, based on the tests conducted on 27 datasets, that the proposed CRNNTL method can outperform or compete with state-of-art methods. Additionally, the paper gives valuable insight in how accuracies of virtual screening techniques can be improved. Moreover, public code availability contributes to the transparency and reproducibility of the research, which is highly beneficial. There are, however, some concerns that need to be addressed:

  1. Convolutional layers are used to detect patterns in sequences. The general idea of convolutional layers is the existence of spatial/sequential dependency of input (e.g. in image analysis each pixel is dependent on the ones surrounding it). Since the model is trained using autoencoder’s latent space, it would be beneficial to examine whether such a dependency exists there.

This could be done by checking if permutations of the latent vector members, applied equally to all data, give similar results. It would also be interesting to analyze whether dense layers could be used instead of convolutional layers. If such an architecture yields equally good or better results, this would indicate that members of the latent vector are independent and advocate against the usage of convolutional layers.

  1. The authors mention they derived model architecture by performing hyperparameter grid search. However, some pieces of information are missing, like:
  • Which data was used for grid search? 
  • Was the cross-validation applied, and if so was it on the train dataset (80% derived by K-means clustering)? 
  • Was the testing dataset kept apart from the hyperparameter fine-tuning to avoid potential overfitting? If so, which evaluation results are based on the testing dataset?

  1. It would be advantageous to mention the size of the latent space produced by different autoencoders. Further to this point, it remained unclear whether the AE models taken from literature (e.g. [6]) are used as pretrained models, or the authors trained them themselves (if so, which data was used for their training)?

  1. In line 415, the authors state "Meanwhile, if the size of the dataset is small, batch size can be less than 128." Please specify what do you mean by “small” and how much “less than 128” would be the optimal value. In other words, the authors ought to present their results in a manner that clearly defines the relationship between dataset size and batch size. This piece of information could not be retrieved even from the source code repository.

  1. In line 405, the authors describe that "Early stopping was performed to avoid overfitting." How was that done? What parameters were monitored and what triggered the stopping criterion? Did you monitor validation loss on the test dataset? Please explain more in the text.

  1. In line 399, the authors say "The kernel sizes of the convolution were 5, 2 and 5. And the number of the produced filters were 15, 30 and 60, respectively." The motivation behind the chosen values remained unclear. Were they a part of the hyperparameter search, or were they taken from the literature?

  1. In Table 4, for Emmax dataset, the results of RFb model should not be bold, as its R2 score is lower than that of the SVM and AugCRNN.

It is the opinion of this reviewer that providing answers and additional explanations regarding these issues would greatly enhance the quality of the paper.

Author Response

Dear reviewer,

We would like to thank you for carefully assessing our paper and providing constructive and valuable critique, which has allowed us to significantly improve our work. Attached is the point-by-point response to you. Questions from you are written in black, and our responses are written in red.

Reviewer 2 Report

The manuscript “CRNNTL: convolutional recurrent neural network and transfer learning for QSAR modeling in organic drug and material discovery” proposes the combination of two types of neural networks (convolutional and recurrent) with transfer learning to overcome limitations such as molecular sequence and deficiency data in computing prediction. The presented work is great and interesting, but several issues must be address before the manuscript could be considered for publication:

- I salute the code availability on GitHub, but some basic details about the working protocol must be included in the Method section (language, packages, software...;
- Data augmentation performed to optimize the models used 9 smiles codes. Some additional details are needed. 9 codes from each set or from all sets? Would that be enough? Why 9 codes?
- In general, we think of small data sets when we hear “data scarcity”, but in some cases it can also refer to the diversity /uniformity of the data. In the present paper, other small sets e.g. Beetox (188 compounds) or JAK3 (868 compounds) have higher ROC-AUC values (0.92 and 0.77 respectively) compared to the 0.72 of the SIDER dataset (1412 compounds). A similar example could be found for regression datasets: LogS (1144 compounds) gives an r2 value of 0.93 compared to the 0.5 of PI3 dataset (2995 compounds). I agree that the data number influences the results, but the diversity/uniformity of the data may do the same. Is there any way to illustrate the diversity of the data in a set? Can diversity (or lack thereof) influence the results? 
-  Table 7 should also include datasets used for modeling the material properties.
- Does the BP dataset contain isomers? It is much larger than MP dataset and with better results. 
- Were different validation methods used to model drug and material properties due to size differences between datasets?
- Intuitively, transfer learning has a higher success rate when there is a similarity between the modeled property and the one on which the transfer is made. Is it not very clear, the transfer for the MP set was made from BP or all the sets taken as source?

And a question outside the review:
- If isomeric smiles are used, could the method be extended to stereoisomers, not just constitutional isomers?

Author Response

Dear reviewer,

We would like to thank you for carefully assessing our paper and providing constructive and valuable critique, which has allowed us to significantly improve our work. Attached is the point-by-point response to you. Questions from you are written in black, and our responses are written in red.

This manuscript is a resubmission of an earlier submission. The following is a list of the peer review reports and author responses from that submission.